# Electron shelving of a superconducting artificial atom

Nathanaël Cottet[1,2,3], Haonan Xiong [1,3], Long B. Nguyen [1], Yen-Hsiang Lin[1] & Vladimir E. Manucharyan [1✉]

Interfacing long-lived qubits with propagating photons is a fundamental challenge in quantum technology. Cavity and circuit quantum electrodynamics (cQED) architectures rely on an off-resonant cavity, which blocks the qubit emission and enables a quantum non-demolition (QND) dispersive readout. However, no such buffer mode is necessary for controlling a large class of three-level systems that combine a metastable qubit transition with a bright cycling transition, using the electron shelving effect. Here we demonstrate shelving of a circuit atom, fluxonium, placed inside a microwave waveguide. With no cavity modes in the setup, the qubit coherence time exceeds 50 μs, and the cycling transition's radiative lifetime is under 100 ns. By detecting a homodyne fluorescence signal from the cycling transition, we implement a QND readout of the qubit and account for readout errors using a minimal optical pumping model. Our result establishes a resource-efficient (cavityless) alternative to cQED for controlling superconducting qubits.

[1] Physics Department, University of Maryland, College Park, MD 20742, United States. [2] Université Lyon, ENS de Lyon, Université Claude Bernard Lyon 1, CNRS, Laboratoire de Physique, F-69342 Lyon, France. [3] These authors contributed equally: Nathanaël Cottet, Haonan Xiong. ✉email: vmanuchar@gmail.com

A rtificial circuit atoms interact with radiation much more strongly than conventional atoms[1,2]. In particular, a nearly unit efficiency scattering of single photons was observed with a superconducting flux qubit inductively coupled to a planar transmission line[3], followed by a number of fundamental demonstrations involving resonant fluorescence of transmon qubits[4–7]. However, the lack of diversity among transition frequencies and selection rules in those two popular circuit atoms prevents their full-blown quantum control without invoking an extra cavity mode. In fact, indirectly interfacing superconducting qubits with radiation via far-detuned cavity resonators has been the sole option over the last decade to achieve both a state-of-the-art coherence time and a quantum non-demolition (QND) readout[8]. By contrast, a large portion of optical quantum technology relies on atoms that combine two types of transitions: a dark one with a long coherence time to store a qubit, and a bright one, which can fluoresce conditioned on the qubit state[9]. This effect of conditional fluorescence is known in atomic physics as electron shelving[10] and it enables a direct interface between high-coherence qubits and propagating photons[11]. Inspired by this observation, we ask the following question: can we eliminate the cavity in a cQED setup by switching to three-level circuit atoms with appropriately designed transitions? Besides facilitating quantum communication and distributed quantum computing[12–14], having a more direct qubit-photon interface may duly free up the on-chip space in quantum processor circuits, the majority of which is usually occupied by the readout resonators.

In this work we describe the electron shelving effect in a multilevel circuit atom—fluxonium[15–18]. Specifically, we demonstrate a conditional fluorescence readout of a qubit, which is directly coupled to a coaxial cable of the measurement apparatus, and yet has a relatively high coherence time. Fluxonium circuit consists of a Josephson junction shunted by a high-value inductance (superinductance) and a capacitance, here in the form of a bow-tie antenna (Fig. 1a). When cooled-down near 10 mK in a dilution refrigerator, the circuit dynamics can be characterized by a single collective degree of freedom, the superconducting phase-difference $\phi$ across the junction. At half-integer magnetic flux through the loop, $\phi_{ext} = \pi$, the variable $\phi$ behaves like the coordinate of a particle inside a symmetric double-well potential and the spectrum is first-order insensitive to flux noise (Fig. 1b). Tunneling across the barrier forms the qubit states $|0\rangle$ and $|1\rangle$,

split by a relatively low-frequency $\omega_{01}$ (in the present device $\omega_{01} = 2\pi \times 1.152$ GHz). The non-computational states $|2\rangle$, $|3\rangle$, etc., are the orbitals at energy above the barrier. Higher-frequency transitions $|1\rangle - |2\rangle$ and $|0\rangle - |3\rangle$ are natural candidates for cycling, and for technical reasons, we use the latter one at frequency $\omega_{03} = 2\pi \times 6.544$ GHz. Note, transition $|0\rangle - |3\rangle$ is allowed thanks to the absence of the harmonic ladder selection rule in fluxoniums.

Shelving requirements are met by the combination of (i) octaves of frequency separation between the qubit and cycling transitions, (ii) engineered frequency-dependent coupling of the atom circuit to external radiation, and (iii) parity selection rule at $\phi_{ext} = \pi$. We mount the fluxonium chip inside a copper enclosure (not a cavity resonator) with a specially designed input/output port. The port adapts a transverse-electromagnetic (TEM) field in the externally connected coaxial cable to a transverse electric (TE) field inside the enclosure, which in turn couples to the qubit antenna. Such a wireless link[19] between an on-chip circuit and external radiation has a low-frequency cutoff associated with the TE-mode, which we designed to be near the frequency $\omega_{03}$. Therefore, the radiative lifetime of state $|3\rangle$ can be orders of magnitude shorter than that of state $|1\rangle$ even though transitions $|0\rangle - |1\rangle$ and $|0\rangle - |3\rangle$ have comparable matrix elements of the Cooper pair number operator $-i\partial_\phi$ in the present device (Fig. 1d). The radiative decay $|3\rangle \rightarrow |2\rangle$ is similarly suppressed by the TE-mode cutoff, while the direct decay $|3\rangle \rightarrow |1\rangle$ is dipole-forbidden by the parity selection rule. As a result, the driving of transition $|0\rangle - |3\rangle$ would generate fluorescence if and only if the atom (and hence the qubit) starts in state $|0\rangle$. Despite a periodic transit of population through state $|3\rangle$, the circuit rapidly relaxes back to state $|0\rangle$ on switching off the drive. In state $|1\rangle$ the circuit is unresponsive to the drive (shelved). Thus, the conditional fluorescence readout has a built-in QND property, as, ideally, it does not switch the qubit state.

Following earlier work, we monitor fluorescence in the microwave domain using a phase-sensitive (homodyne) scheme[3,4]. To understand the measurement outcome in our single-port setup, let us neglect, for the time being, the possibility of transitions outside the cycling manifold $\{|0\rangle, |3\rangle\}$, and assume the environment to be at zero temperature. According to the input/output theory, the complex amplitude of the outgoing reflected wave $\alpha_{out}$ is given by the sum of the incoming wave amplitude $\alpha_{in}$ and the fluorescence field emitted by the atom[20].

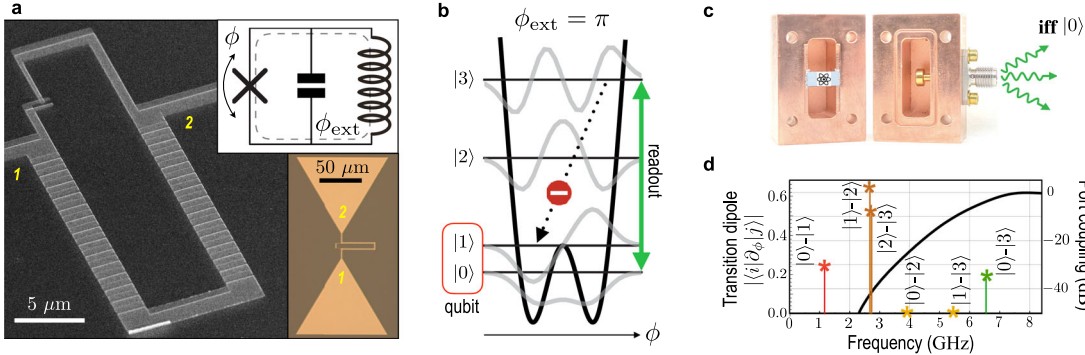

**Fig. 1 Cavityless cQED with fluxonium qubits. a** Electron microscope image of fluxonium's superconducting loop, made of one smaller-area Josephson junction and an array of larger-area junctions. Top inset: fluxonium's effective three elements circuit model. Bottom inset: the smaller-area junction is connected to bow-tie antenna capacitance (optical image). **b** Effective potential seen by the phase-difference $\phi$ across the smaller-area junction (black), energy levels (horizontal black lines), and wavefunctions (gray) at the half-integer external flux through the loop, $\phi_{ext} = \pi$. The qubit levels are $|0\rangle$ and $|1\rangle$, the readout transition is $|0\rangle - |3\rangle$, and even transitions are dipole-forbidden. **c** Setup for a broadband wireless coupling of fluxonium circuit to an external coaxial cable (not shown). **d** Transition frequencies and corresponding matrix elements of the Cooper pair number operator $-i\partial_\phi$ (colored stars). The solid line shows microwave transmission through the enclosure measured using a symmetric two-port configuration. Single-port configuration is used for all other measurements, as shown in Fig. 1c.

For a drive near the readout frequency $\omega_{03}$, this relation takes the simple expression $\alpha_{\text{out}} = \alpha_{\text{in}} - \sqrt{\Gamma}\langle|0\rangle\langle 3|\rangle$, where $\Gamma$ is the rate of direct radiative decay $|3\rangle \rightarrow |0\rangle$, and the expectation value is taken over the dissipative driven steady state. Because the fluorescence contribution $-\sqrt{\Gamma}\langle|0\rangle\langle 3|\rangle$ is zero if the atom is not initially in state $|0\rangle$, the complex reflection coefficient $r = \alpha_{\text{out}}/\alpha_{\text{in}}$ can be related to the ground state population $p_0$:

$$r = 1 - 2p_0 \times \frac{\Gamma^2/2 - i(\omega - \omega_{03})\Gamma}{\Gamma^2/2 + \Omega^2 + 2(\omega - \omega_{03})^2}. \qquad (1)$$

Here the Rabi rate $\Omega = 2\sqrt{\Gamma}\alpha_{\text{in}}$ conveniently represents the readout drive strength in units of [Hz] and $\omega$ is the drive angular frequency. Information about the state of the atom is carried by the fluorescence signal conditionally generated during readout. Its power is proportional to the square of $\Omega|1 - r(\omega, \Omega)|$. This quantity is maximal for $\Omega = \Gamma/\sqrt{2}$ and $\omega = \omega_{03}$, in which case Eq. (1) simplifies to $r = 1 - p_0$. The maximal fluorescence power condition reflects the fact that a linear detection relies on the drive-induced coherence between states $|0\rangle$ and $|3\rangle$, rather than on the population of state $|3\rangle$ in the more common case of photodetection.

## Results

We begin our experiment by characterizing the fluorescence of transition $|0\rangle - |3\rangle$ (Fig. 2a, b). The measured reflection amplitude $r$ as a function of drive frequency $\omega$ and amplitude $\Omega$ fits well to the Eq. (1) model. Only four adjustable parameters are used in the global fit of over ten curves: $\omega_{03} = 2\pi \times 6.544$ GHz, $\Gamma = 2\pi \times 1.74$ MHz, the thermal equilibrium ground state population $p_0^{\text{th}} = 0.78$, and the scaling factor converting the value of $\Omega$ to the RF-source amplitude. In order to ensure that the atom is in the driven steady-state while fluorescence is being integrated, we turn on the drive, wait for a time $\tau_w = 0.5$ μs, then integrate the reflection signal during a time $\tau_m = 0.5$ μs. The wait time is chosen so that it is much longer than the characteristic emission time $1/\Gamma = 91$ ns but much shorter than the time scale of non-radiative processes in the system. At low power, $\Omega \ll \Gamma$, the reflection amplitude $r$ as a function of frequency becomes a circle in the parametric IQ-plot, bounded by the values $\text{Re}[r] = 1$ for $|\omega - \omega_{03}| \gg \Gamma$ and $\text{Re}[r] = 1 - 2p_0^{\text{th}}$ at $\omega = \omega_{03}$. As stronger driving saturates the $|0\rangle - |3\rangle$ transition, the circle deforms into an ellipse and progressively shrinks into a point at $r = 1$ for $\Omega \gg \Gamma$. Agreement between spectroscopy data and the Eq. (1) model indicates that transition $|0\rangle - |3\rangle$ is coupled to itinerant radiation with a nearly unit efficiency, and non-radiative processes are negligible.

The readout characterization presented in Fig. 2a, b provides absolute calibration of the measurement records in terms of the fluxonium's ground state population $p_0$. This property is in contrast with a standard dispersive readout, where the

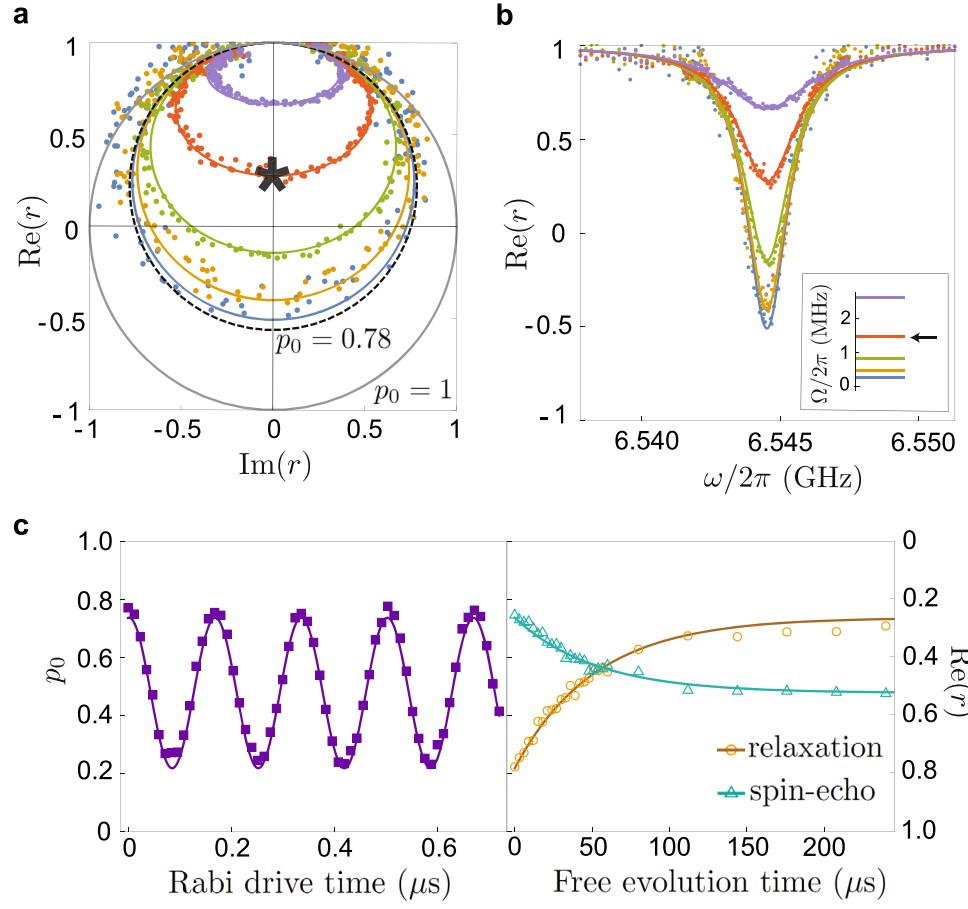

**Fig. 2 Conditional fluorescence readout. a** Complex plane representation and **b** real part of the reflection coefficient $r$ as a function of readout drive frequency $\omega$ at various powers represented by the $|0\rangle - |3\rangle$ transition Rabi frequency $\Omega$ (inset). The experimental data (colored dots) are fitted by the theory of Eq. (1) (lines) with $\Gamma = 2\pi \times 1.74$ MHz and $p_0 = 0.78$ due to the thermal occupation of excited states. In the limit $\Omega \rightarrow 0$, the data in Fig. 2a becomes a circle with a radius $p_0$ (gray circle, $p_0 = 1$ and dashed circle, $p_0 = 0.78$). The optimal readout settings are highlighted by the black star and arrow (see text below Eq. (1)). **c** Rabi oscillations (left panel), energy relaxation (right panel, yellow circles), and spin-echo coherence decay (green triangles) were measured using conditional fluorescence readout at optimal settings. Fitting to exponential decay yields (coincidentally) $T_1 = T_2 = 52$ μs.

measurement records are only known up to a scaling factor to the qubit population and have to be independently calibrated. In order to account for small thermal fluctuations over the course of the experiment, we repeated the previous calibration over 5 days. We find an average thermal equilibrium value of $p_0^{th} = 0.76 \pm 0.03$, which corresponds to the atom being in thermal equilibrium at an effective temperature $T = 45 \pm 5$ mK, which is indeed commonly encountered in superconducting qubits. In what follows, we operate the readout at the maximal signal point, where $p_0 = 1 - r$ (black star on Fig. 2a).

The protocol for time-domain qubit manipulations matches that of traditional circuit quantum electrodynamics. For example, applying a drive at the qubit frequency $\omega_{01} = 2\pi \times 1.15$ GHz prior to the readout pulse results in Rabi oscillations of $p_0$ as a function of drive duration (Fig. 2c—left panel). Operating the circuit slightly off the sweet-spot at $\phi_{ext}/2\pi = 0.507$, i.e., weakly breaking the parity selection rule, we could produce Rabi oscillations between states $|0\rangle$ and $|2\rangle$ by compensating the vanishing transition dipole with a strong drive at frequency $\omega_{02} = 2\pi \times 3.88$ GHz (see Supplementary Note 5). Importantly, the nonzero decay rates of even transitions resulting from breaking the parity selection rule remain several orders of magnitude smaller than other decoherence processes (see Supplementary Table 1). The reflection coefficient measured during Rabi oscillations between states $|0\rangle$ and either $|1\rangle$ or $|2\rangle$ yields equilibrium ratios $p_1^{th}/p_0^{th} = 0.329$ and $p_2^{th}/p_0^{th} = 0.007$ independently of the absolute calibration of $p_0^{th}$. Assuming $p_0^{th} + p_1^{th} + p_2^{th} = 1$, we obtain the equilibrium value of $p_0^{th} = 0.75$, in close agreement with the spectroscopic calibration (see Supplementary Note 7). We checked that the qubit can be initialized with $p_0 \approx 0.9$ by strongly driving the $|1\rangle - |3\rangle$ transition. During such a drive, entropy is removed from the on-chip circuit via spontaneous emission of photons $|3\rangle \rightarrow |0\rangle$ into the measurement line (see Supplementary Note 6). However, the value $p_0^{th}$ turns out to be sufficiently high for the purpose of our time-domain experiments and hence we performed them all starting from thermal equilibrium.

The energy relaxation signal following a $\pi$-pulse at frequency $\omega_{01}$ is exponential with a characteristic decay time $T_1 = 52$ μs, which is nearly three orders of magnitude longer than the radiative decay time of state $|3\rangle$ (Fig. 2c—right panel). Yet, given the highly suppressed port coupling at the qubit frequency, such value of $T_1$ is still too short to be explained by radiative decay. In fact, the qubit decay is likely due to the dielectric loss in the circuit's antenna capacitance with an effective quality factor $Q_{diel} = 5 \times 10^5$. The spin-echo signal decay is also exponential with a characteristic coherence time $T_2 = 52$ μs. The additional dephasing rate $\Gamma_\varphi = 1/T_2 - 1/2T_1$ can be explained by a thermal occupation of the propagating modes near the frequency $\omega_{03}$, which would induce upward jumps $|0\rangle \rightarrow |3\rangle$. Assuming the rate of such jumps is $\Gamma_\varphi$, it is related to the effective photon temperature $T$ at the microwave port via the detailed balance equation $\Gamma_\varphi = \Gamma \exp(-\hbar\omega_{03}/k_B T)$ (see Supplementary Note 8). The value $T = 45 \pm 5$ mK, previously extracted from the equilibrium population measurement, indeed accounts for the extracted value of $\Gamma_\varphi$. This dephasing channel can be exponentially suppressed in better-thermalized setups.

Finally, we explore the QND character of the conditional fluorescence measurement. An elementary test consists of measuring the atom population dynamics induced by the readout process. To do so, we drive transition $|0\rangle - |3\rangle$ for a duration $\tau \gg 1/\Gamma$, pause for a brief period for state $|3\rangle$ to relax, swap the state of interest with state $|0\rangle$, and perform our conditional fluorescence readout (Fig. 3a—top panel). As a result, we acquire the population transients $p_0(t)$, $p_1(t)$, and $p_2(t)$ of states $|0\rangle$, $|1\rangle$, and $|2\rangle$, respectively, during a drive at readout frequency. The fidelity of this population measurement is limited by the swap gate fidelities, which is above 99% for $|0\rangle - |1\rangle$ and above 90% for $|0\rangle - |2\rangle$, measured using randomized benchmarking (see Supplementary Note 3). The $p_0(t)$-data indicates a gradual leakage of the population outside the cycling manifold $\{|0\rangle, |3\rangle\}$ on a characteristic time scale $\tau_{cyc} = 9.6$ μs. Importantly, the leakage involves only states $|1\rangle$ and $|2\rangle$ as the data satisfies $p_0 + p_1 + p_2 = 1$ within the measurement uncertainty (Fig. 3a— bottom panel). The average number of fluorescence cycles, in our case $N_{cyc} = \Gamma \times \tau_{cyc} \approx 105$, is usually used in optical shelving to quantify the deviation of the readout from ideal QND behavior.

Fluorescence lifetime $\tau_{cyc}$ can be accounted for by non-radiative transitions outside the cycling manifold. We further constrain the energy relaxation rates in our circuit by measuring populations of states $|0\rangle$, $|1\rangle$, and $|2\rangle$ as a function of time following a $|0\rangle - |2\rangle$ swap (Fig. 3b). Combining the two data sets shown in Fig. 3a and b, we construct a relaxation model based on two additional decay mechanisms, dielectric loss and quasiparticle tunneling across the junction. The model adequately matches the six experimental curves with only three adjustable parameters: the dielectric loss quality factor $Q_{diel} = 5 \times 10^5$, previously introduced to account for decay $|1\rangle \rightarrow |0\rangle$, the dimensionless quasiparticle density $x_{qp} = 10^{-6}$, and a fudge-factor of 2.5 in front of the $|3\rangle \rightarrow |2\rangle$ decay rate. The latter could be due to a nearby two-level fluctuator[21]. This model highlights two specific fluorescence-blocking processes. A small role is played by the direct parity-breaking decay $|3\rangle \rightarrow |1\rangle$ with a characteristic time 36 μs due to quasiparticle tunneling. The dominant readout error, though, comes from the decay $|3\rangle \rightarrow |2\rangle$ with a characteristic time of 3.6 μs due to dielectric loss in the fluxonium capacitance (see Supplementary Note 8).

## Discussion

In summary, we showed that a single circuit degree of freedom can combine both the qubit and the readout functions. The readout was made possible by the phenomenon of conditional fluorescence, analogous to electron shelving of atomic clocks. The required pair of dark and bright transitions arises naturally from fluxonium's strong anharmonicity, in contrast to more complex circuit implementations of $V$- and $\Lambda$-type systems[22–24]. In future experiments, one can use transition $|1\rangle - |2\rangle$ for cycling, in which case the fluorescence lifetime $\tau_{cyc}$ should increase to the $T_1$-limit. The 3D coupling setup in Fig. 1c can be replaced by the more common capacitive connection to an on-chip transmission line. In that case, the qubit emission would still be strongly suppressed at low frequencies[16], approximately as $1/\omega_{01}^3$ for $\hbar\omega_{01} \gtrsim k_B T$. Finally, adding a parametric amplifier would enable a single-shot readout even at the single-photon power level[25] (see Supplementary Note 4).

Going cavityless offers the most direct interface between superconducting qubits and propagating microwave photons without degrading the qubit coherence time. Such an interface can be useful for exploring quantum optics in the microwave range, with new applications, such as photon routing controlled by superposition of states[26,27]. In the context of waveguide QED, our setup implements a quantum node which can act simultaneously as an emitter and as a memory. This development may pave the way to the preparation of highly-entangled states over a quantum network[14] and their use for distributed quantum computation protocols, such as teleported gates[28]. Of particular interest is to combine the shelving readout with recently developed single microwave photon counters[29–31]. Last but not least, eliminating readout resonators from large-scale quantum processor circuits may save precious on-chip space.

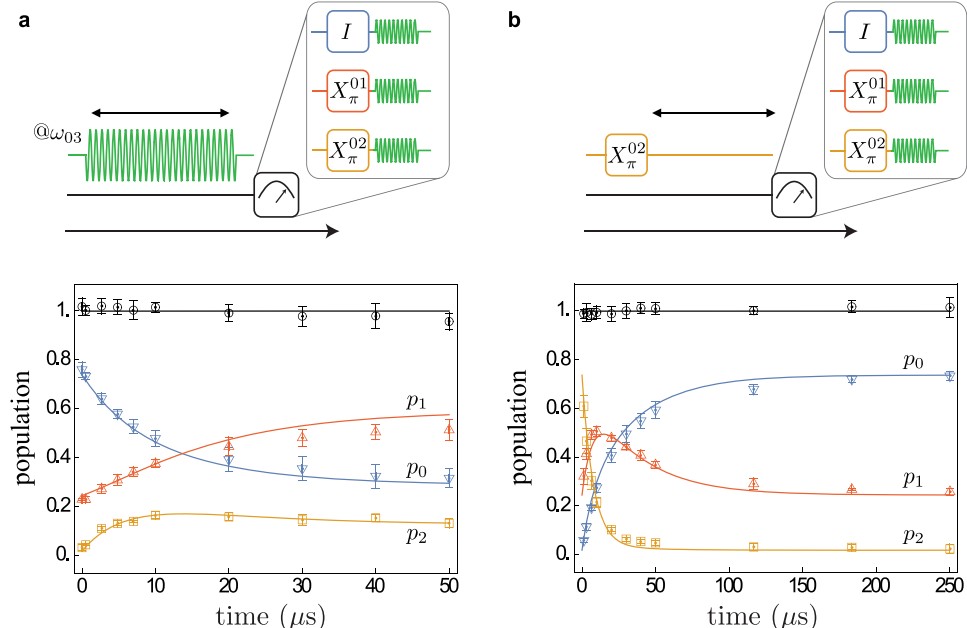

**Fig. 3 Transient dynamics of fluorescence. a** Time-domain evolution of populations $p_0$ (down blue triangles), $p_1$ (up red triangles), and $p_2$ (yellow squares) induced by the cycling of the readout transition (pulse sequence on top). Note that $p_0$ decays approximately exponentially in time with a characteristic time scale $\tau_{cyc} = 9.6\,\mu s$, after which the population is transferred from state $|0\rangle$ to states $|1\rangle$ and $|2\rangle$. The black circles indicate $p_0 + p_1 + p_2$ and the error bars represent the standard deviation coming from the repetition of the measurements over two days. **b** Free evolution of populations starting predominantly from state $|2\rangle$ (pulse sequence on top). The data in both (**a**) and (**b**) is adequately explained by an optical pumping model (solid lines) (see text) where the dominant error mechanism is a non-radiative decay due to dielectric loss.

## Methods

**Derivation of Eq. 1.** A drive at angular frequency $\omega$ near readout frequency induces damped Rabi oscillations of the readout transition with the Hamiltonian

$$H/\hbar = \frac{(\omega - \omega_{03})}{2}(|3\rangle\langle 3| - |0\rangle\langle 0|) - i\frac{\Omega}{2}(|3\rangle\langle 0| - |0\rangle\langle 3|) \quad (2)$$

and $\Omega = 2\sqrt{\Gamma}\alpha_{in}$. In all generality, one should consider all sources of loss and dephasing such as spontaneous emission in the line, non-radiative decay, and extra-dephasing due to flux noise, including transitions from the readout subspace to other states of the atom. However, the readout transition has been designed so that its emission rate in the line overcomes by many orders of magnitude the other decay and dephasing processes. Assuming an environment at zero temperature, the density matrix of the atom $\rho$ evolves according to the following Lindblad equation

$$\dot{\rho} = -\frac{i}{\hbar}[H, \rho] + \Gamma \mathcal{D}[|0\rangle\langle 3|](\rho) \quad (3)$$

where $\mathcal{D}$ is the Lindblad superoperator. The steady-state of this equation yields to the expectation value

$$\langle |0\rangle\langle 3| \rangle = \frac{\Omega}{\Gamma} \frac{\Gamma^2/2 - i(\omega - \omega_{03})\Gamma}{\Gamma^2/2 + \Omega^2 + 2(\omega - \omega_{03})^2}, \quad (4)$$

from which we derive Eq. (1). The good agreement between the theoretical model and the experimental data validates the assumptions made in the previous derivation.

**Reflection coefficient calibration.** The measured reflection coefficient is only known up to a scaling factor and has to be calibrated. In fact, the readout drive undergoes several stages of attenuation before reaching the artificial atom, while the fluorescence signal is amplified before digitization. Calibration is performed as follows. The circuit is flux biased at $\phi_{ext} = 0$ and is probed in reflection between 6.3 and 6.6 GHz. The external flux is chosen so that no circuit transition exists in the frequency band of interest. The acquired signal $s_{cal}(\omega)$ therefore accounts for the temperature-dependent attenuation and filtering of the lines as well as the whole acquisition setup amplification chain. Once the atom is biased at the wanted flux, the reflection coefficient $r$ is deduced from the experimentally measured signal $s_{exp}(\omega)$ by

$$r = \frac{s_{exp}(\omega)}{s_{cal}(\omega)} 10^{(P_{cal} - P_{exp})/20} \quad (5)$$

with $P_{cal/exp}$ the room-temperature power of the readout drive during the calibration/experiment, expressed in dBm.

**Relaxation model of transient dynamics.** The relaxation model used to reproduce the data represented in Fig. 3 considers three distinct decay mechanisms. A jump from a state $|i\rangle$ to $|j\rangle$ can occur due to radiative decay in the line, dielectric loss, or quasiparticle tunneling in the junction, with the rates $\Gamma_{ij}^{rad}$, $\Gamma_{ij}^{diel}$, $\Gamma_{ij}^{qp}$, respectively. All rates are computed assuming an equal bath temperature $T = 50$ mK for all decay mechanisms (see Supplementary Note 8).

We simulate the dynamics of the atom using the Python library QuTiP[32]. The spectrum of the fluxonium hamiltonian $H_f = 4E_C(-i\partial\phi)^2 + E_L\phi^2 - E_J\cos(\phi - \phi_{ext})$ is first computed in a Hilbert space of dimension $100 \times 100$, to ensure accuracy of the computed eigenfrequencies $\omega_i$ and eigenstates $|i\rangle$, $i \leq 99$. We then solve numerically the Lindblad master equation in the Fock state basis for a smaller Hilbert space containing the first ten eigenstates. This takes into account the fact that high-energy states do not contribute to the atom dynamics and allows for a faster computing time. The master equation contains all the jump operators $\{\sqrt{\Gamma_{ij}^{rad} + \Gamma_{ij}^{diel} + \Gamma_{ij}^{qp}}|j\rangle\langle i|\}_{i,j \leq 9}$ computed from the diagonalization of $H_f$. The initial state of the atom is the thermal equilibrium state $\rho_{th}$ for Fig. 3a and a perfect swap between $|0\rangle$ and $|2\rangle$ on $\rho_{th}$ for Fig. 3b.

The Hamiltonian for Fig. 3a is $H = -i\frac{\Omega}{2}(|3\rangle\langle 0| - |0\rangle\langle 3|)$ and represents the readout drive, while it is 0 for Fig. 3b. They correspond to computing the evolution in the frame rotating at all the eigenfrequencies, i.e., applying the unitary $U(t) = \exp(-iH_f t/\hbar) = \exp(-i\sum_j \omega_j |j\rangle\langle j| t)$ to the driven and undriven hamiltonians, respectively. We find that the transient dynamics of fluorescence is best reproduced for $\Omega = 0.95\Gamma/\sqrt{2}$. This deviation from the ideal value can be due to small drifts between the measurements of Figs. 2 and 3.

## Data availability

The data used in the manuscript is available from the corresponding author on a reasonable request.

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

## Acknowledgements
We thank Ray Mencia for providing samples at the initial stages of this work and Nathan Langford, Shimon Kolkowitz, and Benjamin Huard for useful discussions. We acknowledge funding from Sloan Foundation, NSF-PFC at JQI (1430094), ARO-LPS HiPS (W911NF-18-1-0146), and ARO-MURI (W911NF-15-1-0397).

## Author contributions
H.X. fabricated the device and along with Y.-H.L. acquired and analyzed the data. Y.-H.L. designed the coaxial-to-waveguide adapter and built the low-temperature measurement setup. L.B.N. contributed to circuit design, room-temperature instrumentation, and to identifying decoherence mechanisms. N.C. performed initial conditional fluorescence experiments and developed procedures for analyzing and modeling the data. V.E.M. managed the project.

## Competing interests
The authors declare no competing interests.
