## [Peer Review File · Nature Communications]

REVIEWERS' COMMENTS

Reviewer #1 (Remarks to the Author):

The authors responded properly to the questions by the Reviewers and revised the manuscript accordingly. I recommend it for publication in *Nature Communications*.

Reviewer #2 (Remarks to the Author):

The authors have have addressed most of my comments in the revised manuscript and its supplementary material. However, I do not quite agree with the language used to describe "cavityless" operation. The experiment makes use of a non-resonant, but highly dispersive coaxial-to-waveguide adapter where the cut-off frequency is needed to avoid damping of the protected transition. I understand that the statement is made in referring to possible technical benefits in scaling quantum processors but the current presentation somewhat dilutes the role of the frequency-dependent electromagnetic environment in implementing electron shelving. I recommend some refinements in the language, but otherwise I recommend the manuscript for publication.

Reviewer #3 (Remarks to the Author):

First I would like to thank the authors for their detailed answer to my previous comments. I am also glad that they significantly rewrote their manuscript and agreed to mention previous shelving readout schemes implemented with superconducting circuits (references 29 to 31) and more specifically the work of Mineev et al. which reports, to the best of my knowledge, the first realization of such scheme in cQED.

Now I would like to comment on what seems to be the main disagreement between the authors and myself. Shelving literally means “putting on a shelf” or “putting aside”. Hence shelving readout requires only a combination of bright and dark transitions so that the computational one is isolated during read-out. Originally, it is true, this scheme was based on conditional fluorescence but I don’t think that this is a requirement to call readout a shelving one. As a consequence I am still convinced that their work does not provide the first experimental demonstration of “electron shelving” in a circuit.

Also conditional fluorescence was already reported many times in cQED (references 16 to 18 or Naghiloo, M., Foroozani, N., Tan, D. et al. Mapping quantum state dynamics in spontaneous emission. *Nat Commun* 7, 11527 (2016)).

Then regarding novelty, the genuinely new finding is a cavity-less and efficient readout scheme as claimed by the authors.

As a referee, I am also required to comment on the impact of the work. Here I would like to point two major limitations of the readout scheme described by the authors.

First it is not QND at all time. And this is an intrinsic limitation, not a technical one that can be overcome easily. In their reply to my previous comment they claim that this is well known and that this is a textbook notion in atomic physics. I guess it is probably true for readers coming from this community, probably less for physicists working with superconducting qubits. This has strong implications: contrary to dispersive readout, the shelving scheme presented here prevents any continuous measurement.

Second, conditional fluorescence implies readout powers in the single photon per inverse bandwidth regime. This is two orders of magnitude lower than what was reported for dispersive readout (see e.g. Sank, D. et al. Measurement-Induced State Transitions in a Superconducting Qubit: Beyond the Rotating Wave Approximation. *Phys Rev Lett* 117, 190503–6 (2016) or Gusenkova, D. et al. Quantum non-demolition dispersive readout of a superconducting artificial atom using large photon numbers, arXiv:2009.14785). Again, this is an intrinsic limitation and this will probably discourage the wide adoption of this measurement scheme. Indeed, for most of the applications, Signal to Noise Ratio (SNR) is the figure of merit to be optimized and since the noise cannot be lowered below the quantum limit, a signal in the single photon regime is a strong disadvantage.

I would be happy to be proven wrong but I doubt that getting rid of a cavity is attractive enough compared to these two limitations.

To conclude, I still believe that the work by Cottet et al. is interesting and sound. The idea to replace a band pass filter (a cavity) by a high pass one (a waveguide with a low frequency cut-off) is elegant. This is for sure interesting and worth exploring but I don’t think that it qualifies this work for a

publication in Nature Communications since the authors don't demonstrate any new physical effect nor a scheme with obvious technical advantage over already existing ones. Then I would suggest an applied physics journal to report this technical development.

Reviewer #1 (Remarks to the Author):

The authors responded properly to the questions by the Reviewers and revised the manuscript accordingly. I recommend it for publication in Nature Communications.

We thank Reviewer #1 for their comments and for the recommendation to publish.

Reviewer #2 (Remarks to the Author):

I recommend some refinements in the language, but otherwise I recommend the manuscript for publication.

We thank Reviewer #2 for their comments and for the recommendation to publish.

I do not quite agree with the language used to describe "cavityless" operation. The experiment makes use of a non-resonant, but highly dispersive coaxial-to-waveguide adapter where the cut-off frequency is needed to avoid damping of the protected transition. I understand that the statement is made in referring to possible technical benefits in scaling quantum processors but the current presentation somewhat dilutes the role of the frequency-dependent electromagnetic environment in implementing electron shelving.

Actually, the strong dispersion of the coaxial-to-waveguide adapter is generally not necessary for shelving, thanks to the large separation of fluxonium qubit and cycling transition frequencies. It can be replaced by the commonly used capacitive coupling. A simple capacitor too does not transmit at low frequencies and hence can be viewed as a highly dispersive coupler. In case of a fluxonium capacitively coupled to a 50 ohm transmission line, the emission time scales with frequency as $1/f^3$ [see "Superinductance" thesis by V. Manucharyan], and this is plenty to have 3 orders of magnitude of decoupling between cycling and qubit transitions.

To address this useful remark of Reviewer #2 we have implemented the following changes.

1) The first half of paragraph #3 starting with "Shelving requirements were met ..." was revised to more clearly explain the role of each feature of our setup in enabling the shelving readout.

2) In the penultimate paragraph starting with "In summary, ..." we have added the following text: "The 3D coupling setup in Fig. 1c can be replaced by the more common capacitive connection to an on-chip transmission line. In that case the qubit emission would still be strongly suppressed at low frequencies~\cite{manucharyan2012evidence}, approximately as $1/\omega_{01}^3$ for $\hbar\omega_{01} \gtrsim k_B T$."

Regarding the "cavityless" terminology, an electromagnetic vacuum is generally described as a collection of densely spaced cavity modes, so if we stretch the Reviewer's logic, the free space cannot be called a "cavityless" system either. We use the term "cavityless" in our paper as a synonym of absence of discrete electromagnetic modes.

Reviewer #3 (Remarks to the Author):

First I would like to thank the authors for their detailed answer to my previous comments. I am also glad that they significantly rewrote their manuscript and agreed to mention previous shelving readout schemes implemented with superconducting circuits (references 29 to 31) and more specifically the work of Mineev et al. which reports, to the best of my knowledge, the first realization of such scheme in cQED.

We disagree with the Reviewer's claim that shelving readout has ever been demonstrated prior to our work. The work of Mineev et al. relied on coupling one qubit to another qubit and reading out that second qubit dispersively using a cavity. Our work is the first demonstration of a QND qubit readout directly coupled to an electromagnetic continuum.

Now I would like to comment on what seems to be the main disagreement between the authors and myself. Shelving literally means "putting on a shelf" or "putting aside". Hence shelving readout requires only a combination of bright and dark transitions so that the computational one is isolated during read-out. Originally, it is true, this scheme was based on conditional fluorescence but I don't think that this is a requirement to call readout a shelving one. As a consequence I am still convinced that their work does not provide the first experimental demonstration of "electron shelving" in a circuit.

Also conditional fluorescence was already reported many times in cQED (references 16 to 18 or Naghiloo, M., Foroozani, N., Tan, D. et al. Mapping quantum state dynamics in spontaneous emission. Nat Commun 7, 11527 (2016)).

Then regarding novelty, the genuinely new finding is a cavity-less and efficient readout scheme as claimed by the authors.

In our opinion, it does not matter what "shelving" means literally, the meaning of this term is defined by the context. In atomic physics, shelving is a scheme to read out a high-coherence qubit placed in an open electromagnetic environment (3D vacuum). Such a capability has never been demonstrated with superconducting qubits. There is a key statement summarizing the state of the field in our introduction "**In fact, indirectly interfacing superconducting qubits with radiation via far-detuned cavity resonators has been the sole option over the last decade to achieve both a state-of-the-art coherence time and a quantum non-demolition (QND) readout**". Previous work cited by the Reviewer does not contradict this statement on the state of the field. The present one does.

As a referee, I am also required to comment on the impact of the work. Here I would like to point two major limitations of the readout scheme described by the authors.

First it is not QND at all time. And this is an intrinsic limitation, not a technical one that can be overcome easily. In their reply to my previous comment they claim that this is well known and that this is a textbook notion in atomic physics. I guess it is probably true for readers coming from this community, probably less for physicists working with superconducting qubits. This has strong implications: contrary to dispersive readout, the shelving scheme presented here prevents any continuous measurement.

The readout is QND each time it is fired, as described in the manuscript. However, our conditional fluorescence measurement is a strong quantum measurement, unlike the dispersive readout, which is a weak measurement == continuous measurement. Consequently, the shelving readout cannot be used for continuous measurements, as the Reviewer correctly

points out. Is this a limiting factor for quantum computing? For example, the entire ion trap quantum computing platform is based on the fluorescence readout.

Second, conditional fluorescence implies readout powers in the single photon per inverse bandwidth regime. This is two orders of magnitude lower than what was reported for dispersive readout (see e.g. Sank, D. et al. Measurement-Induced State Transitions in a Superconducting Qubit: Beyond the Rotating Wave Approximation. Phys Rev Lett 117, 190503–6 (2016) or Gusenkova, D. et al. Quantum non-demolition dispersive readout of a superconducting artificial atom using large photon numbers, arXiv:2009.14785). Again, this is an intrinsic limitation and this will probably discourage the wide adoption of this measurement scheme. Indeed, for most of the applications, Signal to Noise Ratio (SNR) is the figure of merit to be optimized and since the noise cannot be lowered below the quantum limit, a signal in the single photon regime is a strong disadvantage.

I would be happy to be proven wrong but I doubt that getting rid of a cavity is attractive enough compared to these two limitations.

Indeed, a dispersive readout can operate at a higher power level and hence lead to a higher SNR. However, as we explained in the previous reply: 1) dispersive readout causes qubit to switch the state (a non-QND behavior) at a non-single-photon power level. This effect is not well understood and it is difficult to model. By contrast, our readout error is well understood, it can be modeled by a simple optical pumping model. 2) Current microwave technology is compatible with a single-shot operation already in the single-photon limit, as demonstrated by numerous cQED measurements.

There is always a tradeoff between complexity and performance. Our demonstration endows the superconducting circuit community with the minimal possible qubit-photon interface. The interface enables high coherence time, a QND readout, and it can operate in the single-shot regime after a minor improvement. Time will tell if this trick will be popular in 10 years from now. Experimental physics community has a tremendous inertia. Traditional cQED at first was considered crazy by many mesoscopic physics experimentalists. For example, the group of J. Martinis at UCSB & Google started using cQED readout only around 2010.

To conclude, I still believe that the work by Cottet et al. is interesting and sound. The idea to replace a band pass filter (a cavity) by a high pass one (a waveguide with a low frequency cut-off) is elegant. This is for sure interesting and worth exploring but I don't think that it qualifies this work for a publication in Nature Communications since the authors don't demonstrate any new physical effect nor a scheme with obvious technical advantage over already existing ones. Then I would suggest an applied physics journal to report this technical development.

We respectfully disagree with the Reviewer's conclusion. No one so far succeeded at placing a superconducting qubit in an open environment without losing neither in coherence time nor in the QND readout capability. As to "since the authors don't demonstrate any new physical effect", let us cite the Referee's own words from 2 paragraphs above: "contrary to dispersive readout, the shelving scheme presented here prevents any continuous measurement." The Reviewer

clearly acknowledges that our readout operates in a conceptually different way from the dispersive measurement (strong vs. weak/continuous). Therefore, a statement that we merely replaced a band-pass filter with a high-pass filter contradicts the Reviewer's own words. The physical novelty of our result can be concisely formulated in the summary paragraph: "In summary, we showed that **a single circuit degree of freedom can combine both the qubit and the readout functions.**"